# A Deep Learning based Fast Signed Distance Map Generation

**Zihao WANG** [1,2]                                                                    ZIHAO.WANG@INRIA.FR
[1] *Inria Sophia Antipolis, Nice, France*
[2] *Université Côte d'Azur, Nice, France*

**Clair Vandersteen** [3,2]
[3] *Head and Neck University Institute, Nice, France*

**Thomas Demarcy** [4]
**Dan Gnansia** [4]
[4] *Oticon Medical, Nice, France*

**Charles Raffaelli** [3,2]
[3] *Department of Radiology, Nice University Hospital, France, Inria, Nice, France*

**Nicolas Guevara** [3,2]
**Hervé Delingette** [1,2]

**Editors:** Accepted at MIDL 2020

## Abstract

Signed distance map (SDM) is a common representation of surfaces in medical image analysis and machine learning. The computational complexity of SDM for 3D parametric shapes is often a bottleneck in many applications, thus limiting their interest. In this paper, we propose a learning based SDM generation neural network which is demonstrated on a tridimensional cochlea shape model parameterized by 4 shape parameters. The proposed SDM Neural Network generates a cochlea signed distance map depending on four input parameters and we show that the deep learning approach leads to a 60 fold improvement in the time of computation compared to more classical SDM generation methods. Therefore, the proposed approach achieves a good trade-off between accuracy and efficiency.
**Keywords:** Signed Distance Map, Deep Learning

## 1. Introduction

A Signed Distance Map (SDM) ((Tsai and Osher, 2003)) is a scalar field $f(\mathbf{x})$ giving the signed distance of each point $\mathbf{x}$ to a given (closed) surface, which mathematically translates into the relation $\|\nabla f\| = 1$. In practise, SDMs are 2D or 3D images storing the distance of each voxel center and are widely used to tackle various problems in computer vision or computer graphics fields. In machine learning, SDMs are useful to encode the probability to belong to a shape through log-odds maps ((Pohl et al., 2006)). For instance, given a surface $\mathcal{S}(\theta_S)$ and a scalar $l_{\text{ref}}$, the probability for a voxel $n$ having position $\mathbf{x}_n$ to belong to the surface can be provided through the SDM $\text{SDM}(\mathcal{S}(\theta_S), \mathbf{x}_n)$ at that voxel as $p(Z_n = 1) = \sigma\left(\frac{\text{SDM}(\mathcal{S}(\theta_S), \mathbf{x}_n)}{l_{\text{ref}}}\right)$ where $\sigma(x)$ is the sigmoid function.

While there exist fast (linear complexity) sweeping methods ((Maurer et al., 2003)) for computing SDM from binary shapes, the naive computation of an SDM from triangular meshes has complexity $O(N n_T)$ where $N$ is the number of image voxels and $n_T$ is the number of triangles describing the shape. An example of a generic computation of SDM from meshes is available in VTK (Quammen et al., 2011; Baerentzen and Aanaes, 2005) through the *vtkImplicitPolyDataDistance* class. Since many algorithms are relying on the SDM generation, it is critical to optimize its computation time in various ways (Jia et al., 2018). In medical image analysis, the naive approach leads to poor performances due to the fact that volumetric images and complex shapes are considered. To improve the performance of the SDM calculation, several authors ((Wu et al., 2014; Roosing et al., 2019)) proposed 2D and 3D SDM computation methods that take advantage of graphics processing units (GPU) in order to accelerate the computation. Yet, there does not exist any generic library for fast computation of SDM on GPU, and the availability of specific GPU at test time is a significant limitation for machine learning applications.

Algorithmic optimizations were proposed by various authors ((Jones et al., 2006)) by adopting hierarchical data structures to reach an $O(N \log n_T)$ complexity. For instance, Complete Distance Field Representation (CDFR) ((Jian Huang et al., 2001)) were introduced with triangles structured into 3D grids cells.

Fast approximations of SDM was proposed in (Wu and Kobbelt, 2003) based on structured piece-wise linear distance approximation. Those approaches often require a significant pre-computation stage that can override their computational benefits at later stage.

Recent works of (Chen and Zhang, 2019) and (Park et al., 2019) developed neural networks for the generation of SDM for various of shapes. They rely on an decoder network that takes as input shape parameters and position, and outputs the SDM at that point. The training of those deep SDFs is based on a continuous regression from random samples involving a clamp loss (Park et al., 2019). Those networks are used for shape inference and are point-based signed distance evaluators (without any convolution operation) rather than being generators of SDM. As discussed later in this paper, this is a major issue for fast generation of large images of signed distance maps.

Despite those prior works, there does not exist any generic and efficient way to compute SDM from a triangular mesh on a grid on CPU resources. In this paper, we propose an alternative method for fast computation of SDM based on Convolutional Neural Network (CNN) which does not rely on the rasterization of mesh triangles and does not require any hardware acceleration at test time. Results showed that our approach reduces the SDM computational time complexity significantly without any significant impact on the accuracy of shape recovery.

## 2. Methods and Evaluation

The cochlea is an organ that transforms sound signals into electrical nerve stimuli to the cortex. Cochlea lesions can lead to hearing loss that can be improved by inserting Cochlear Implant(CI) on patients at a middle stage of the disease. Cochlea shape recovery from images is a pivotal step for CI, and the work of (Demarcy, 2017) is a state-of-art method for cochlea shape analysis which makes a computationally intensive use of SDM computations inside Expectation-Maximization loops.

## 2.1. Cochlea Shape Model and Dataset

We rely on a parametric cochlea shape model that represents the shape variability of the human cochlea. It is represented as a generalized cylinder around a centerline having four shape parameters $a, \alpha, b, \phi$, two of them for the longitudinal (resp. radial) extent of the centerline. To compute the SDM of the shape model, the parametric surface was discretized as triangular meshes whose edge lengths are approximately $0.30\pm0.15$ mm (Demarcy, 2017). The SDM was then generated by using VTK library and the *vtkImplicitPolyDataDistance* class which implements a naive SDM algorithm based on point-to-triangle distance computations.

For training the neural network, we generated a static dataset consisting of 625 ($5\times5\times5\times5$) cochlea SDM datasets of size $50\times50\times60$ by uniformly sampling the 4 deformation parameters within user specified ranges. In addition, we performed random data augmentation, by generating online SDMs during the training stage through a random sampling of the 4 shape parameters.

## 2.2. Signed Distance Map Neural Network

Our SDM Neural Network (SDMNN) is an encoder-decoder network with merged layers, its structure being inspired by the well known U-net (Ronneberger et al., 2015). The SDMNN has the four shape parameters as input and generates as output a $50 \times 50 \times 60$ signed distance map (see Fig. 1).

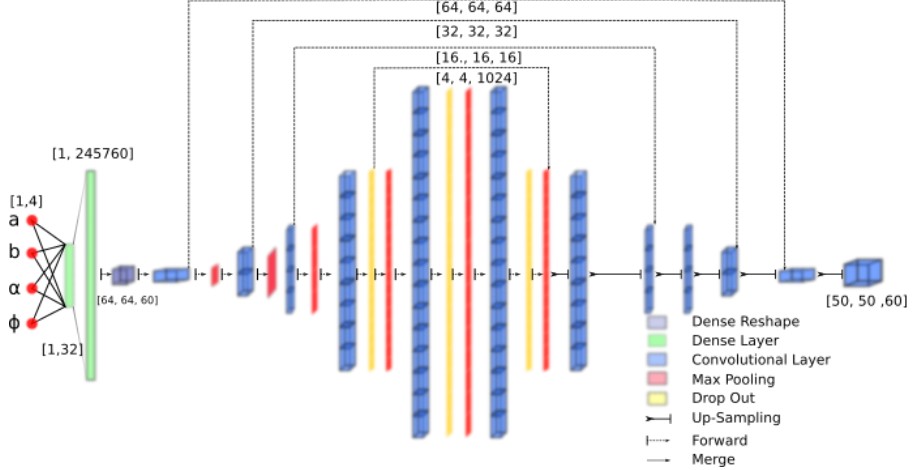

Figure 1: Proposed Signed Distance Map Neural Network (SDMNN)

## 3. Experiments and Evaluation

The SDMNN was trained on one NVIDIA 1080Ti GPU with both static 625 datasets and online random SDMs with a Mean Square Error (MSE) loss for 168 hours. After training, we generated 100 test SDMs with the naive mesh-based VTK code that are associated with random shape parameters. Those were compared to the SDMs generated by the SDMNN for the same shape parameters and the average MSE on the whole images were MSE = $0.006mm$ which is small given that the range of a SDM is $(-0.2mm, 1.3mm)$.

Qualitative results are shown in Fig. 2 (I) where the comparison of the SDMNN and naive mesh-based generated maps is performed by extracting the isocontours associated with the zero (red) and 1mm (yellow) level sets. We see that the isocontours from the SDMNN match closely the ones generated from the mesh. Some small and smooth distorsions appear for the yellow contours. Since in surface reconstruction problems, the main focus of SDM is on the zero level set, the errors of the yellow isocontours are likely not to entail any major reconstruction errors. To verify the accuracy of the zero level isocontour, we have extracted the zero isosurface by the marching cubes algorithm associated with the standard shape values and compared that reconstructed surface with the original triangulated mesh model (the one used to generate the mesh SDM). In Fig. 2 (II) the 2 surfaces are overlaid showing that the SDMNN isosurface is as smooth as the original mesh and that the 2 surfaces are very close indeed.

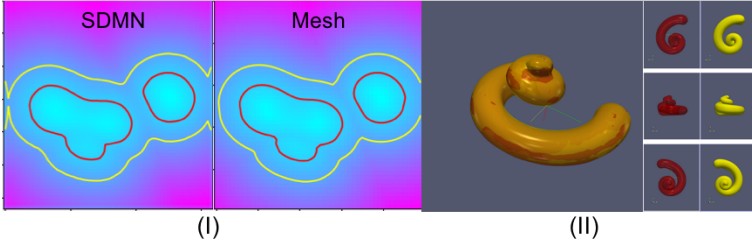

Figure 2: (Left-I) comparison between isocontours extracted from an SDM generated by the SDM neural network (left) and classical method (right);(Right-II) comparison of reconstructed 0-isosurface between the two methods.

The proposed approach is evaluated quantitatively in three ways. First, we compare the computation times between VTK mesh-based SDM generation and the SDMNN-based generation. All evaluations were performed on a Dell Mobile Workstation with Intel(R) Core(TM) i7-7820HQ @ 2.90GHz CPU. We show in Table 1 that the SDM neural network is about 66 times more efficient to generate a SDM than the classical method. Second, the performance was also compared for fitting a cochlea shape model on a clinical CT image as in (Demarcy, 2017) which requires several hundreds of evaluations of signed distance maps. In such case, the speedup was shown to be about 11 times faster than the mesh-based alternative. We also implemented the DeepSDF and IM-NET (Park et al., 2019; Chen and Zhang, 2019) for the generation of SDM of the cochlea with 4 shape parameters. For a fair comparison, we run DeepSDF (which is very similar to the IM-NET) to test its computational efficiency to fill a (60, 50, 50) SDM grid in one batch. The resulting computing time is 28s as shown in Table 1 which is even worse than the default VTK algorithm. This shows that there is high price to pay to have a point-based network rather than a image-based network. Furthermore, we found the accuracy in terms of signed distances of both networks to be significantly worse than our proposed SDMNN.

Thirdly, we evaluated the difference in terms of estimated shape parameters after fitting 9 clinical CT cochlea volumes using both mesh-based and SDMNN methods. This lead to recover the 4 shape parameters $a, \alpha, b, \phi$ on each of the 9 cochleas that are stored in vector $P_{mesh}$ when using mesh-based SDM generation method and vector $P_{SDMNN}$ with SDMNN.

Table 1: Different Methods Computational time for SDM generation (h:m:s)

| Generation Time | SDMNN | Mesh based SDM | DeepSDF |
|---|---|---|---|
| Single SDM | 0:00:00.2 | 0:00:10.7 | 0:00:28.1 |
| Shape Fit | 1:05:02.1 | 12:15:45.4 | Failed |

The errors in shape parameters $P_{err} = \|P_{mesh} - P_{SDMNN}\|$ are reported in Table 2 showing negligible discrepancies given that the parameters magnitude (see head of Table 2).

Table 2: Shape parameters estimation error for SDMNN compared to mesh based SDM

| Parameters Name | a | $\alpha$ | b | $\varphi$ |
|---|---|---|---|---|
| Parameters Range | (2.0, 5.0) | (0.0, 1.2) | (0.05, 0.25) | $(-\pi/4, \pi/4)$ |
| Mean shape parameters errors $P_{err}$ on 9 cases. | 2.06e-08 | 2.53e-08 | 5.4e-08 | 1.00e-09 |

## 4. Conclusion

In this paper, we have proposed a deep learning-based fast signed distance map generation method. We showed quantitatively and qualitatively that it can generate 3D SDM in less than 300 ms, while having an accuracy suitable for shape-recovery, with no noticeable changes in recovered shape parameters. This CNN based SDM generation model can be used for any parametric shape model for SDM generation and does not require any GPGPU resources after training, which is compatible with a clinical environment. While other point-based approaches such as DeepSDF and IM-NET have been also proposed recently, the time overhead to fill a regular grid appears to be fairly large. The current approach is probably suitable only when the number of shape parameters is small since the number of SDMs in the training set should grow quadratically with the number of shape parameters. Future work will look at additional strategies to speed-up the training stage and improve the output accuracy.

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
