# OpenReview forum: "A Deep Learning based Fast Signed Distance Map Generation"
_MIDL.io/2020/Conference — MIDL 2020_

### Official Review · AnonReviewer4 · 2020-03-09
**A Deep Learning based Fast Signed Distance Map Generation**

**Rating:** 1
**Confidence:** 4

**Review:**

Quality:
Study well designed overall.

-Clarity:
Paper clearly written. Fig 1 can be misleading when showing layers stacked vertically.

-Originality:
-- This paper does not deal with medical imaging, but only with shape encoding.

--Two recent works [1,2], not cited by this paper, have proposed a CNN-based solution for computing Signed Distance Functions, one even learning how to represent different classes of shapes [1].

Ref:
[1] Park JJ, Florence P, Straub J, Newcombe R, Lovegrove S. Deepsdf: Learning continuous signed distance functions for shape representation. InProceedings of the IEEE Conference on Computer Vision and Pattern Recognition 2019 (pp. 165-174).
[2] Chen Z, Zhang H. Learning implicit fields for generative shape modeling. InProceedings of the IEEE Conference on Computer Vision and Pattern Recognition 2019 (pp. 5939-5948).

- Significance of this work
No clear gain demonstrated versus existing methods to compute SDF. This is more a "feasibility" study. I do not fully agree that complexity of SDF computing with classic  methods (e.g. fast marching) is "a bottleneck"
Also the cochlear shapes are simple and smooth, modeled as "generalized cylinder around a centerline having four shape parameters". This greatly limits significance.
Finally, SDM of size 50 × 50 × 60 can seem quite small to encode more complex shapes.

-Pros and cons:
Pros
- Clear, well written

cons:
- This paper does not deal with medical imaging, but only with shape encoding.
- Shapes being studied is smooth and simple.

---

### Official Review · AnonReviewer1 · 2020-03-09
**A sound methodology for parametric surface to signed distance maps generation**

**Rating:** 3
**Confidence:** 4

**Review:**

This paper proposes a new U-net like architecture for fast generation of signed distance functions (SDF) from parametric surfaces.  A mapping is learned between surface parameters and the corresponding SDF on synthetic data, and is compared to a vtk class performing conventional SDF calculation using Eikonal solving. The paper is well written and fluent and the methodology is very sound. The authors show a good correspondence between the two methods, with a much lower computational time when inferring SDF using their approach.  The authors show almost perfect correspondence between conventional SDF calculation and their approach on a 9 surface clinical cochlear dataset. One could ask  how the network learned so well with synthetic training examples.  The choice of sampling uniformly over the parameter space seems very suboptimal. In general, it is likely that some parameters have more influence than others on the output shape. I understand this is a short paper and that this could not be discussed.

---

### Official Review · AnonReviewer3 · 2020-03-10
**An original use of neural networks**

**Rating:** 3
**Confidence:** 5

**Review:**

This paper suggests replacing the computationally expensive step of computing distance maps with a neural network. The network is trained to generate a 3D distance map from 4 parameters of a shape model (therefore bypassing the mesh representation).
Neural networks are here mainly considered as fast approximators rather than actual predictors, which is a rather original and interesting approach.

My main remarks:
- The comparison to a naive VTK algorithm is a bit unfair. The computation time of the mesh-based SDM seems particularly high. There are also faster methods that compute an approximation of the SDM.
- Based on the description of the training/validation splitting, there is no guarantee that the validation parameters are not included in the training phase. I am aware that it does not matter that much for the final application (which is just to reproduce a given shape model so there is no real risk of overfitting), but it still biases the evaluation.
- There seems to be some border artifacts visible in Figure 2. (Left-I), which are probably due to some padding operations in internal computations of the U-Net. Usually this is not a major problem since segmented objects are rarely touching the boundary, but here it seems that it does change the topology of the yellow curve so it might be worth fixing.


Should the authors decide to make it a longer paper, here are a couple of ideas that could be considered:
- Here only 4 parameters are considered as an input of the network.
For many applications, this is not enough to get an accurate shape model. It would be interesting to see how well the network is able to approximate the shape model as the number of input parameters increases.
- One of the core properties of distance maps is that their gradient is always one. The deviation of the network output’s gradient to 1 would be another metric worth reporting.
- Taking this idea one step further, this could even be implemented as part of the training loss, so that this property is enforced onto the network.

Minor remarks:
- The width of the first layer 245760 seems a bit arbitrary - can you quickly mention how this was chosen? Also it is a bit weird because it is not resizable to 64 x 64 x 64.
- Please specify the time unit in Table 1.
- Typo L40: “this prior work”
- Typo L41: “resources”
- Typo L95 “recover”
- Typo L101: “a deep learning [...]”

---

### Official Review · AnonReviewer2 · 2020-03-11
**signed distance map**

**Rating:** 2
**Confidence:** 5

**Review:**

The authors propose a neural network approach that aims to generate signed distance functions. They use a U-net type encoder-decoder to expand and then collapse the feature layers as the network architecture. They use the a three dimensional cochlea shape model represented by 4 parameters (representing the longitudinal and the radial extents of the centerline). Their approach leads to a 60 times increase in computation time compared to signed distance function generation methods.

This paper makes a good but very restricted contribution to the generation of signed distance functions. Since the paper used a cochlear dataset that essentially was described by 4 parameters, is the neural network exclusively useful only on such type of data? The generalizability to other signed distance functions is in question.

The authors should provide more evaluation results on the 100 test datasets. For e.g. dice coefficient or the Jacquard index between the signed distance maps would be useful.


The method yields good results on 9 clinical datasets (previously unseen by the model).

In summary, the paper makes a good contribution of learning signed distance function parameters for the cochlear dataset, but its applicability to other datasets has not been demonstrated. I would add that the mapping from a parametric mesh to a signed distance function is more of a topological map than a geometric map. Thus it would be useful to discuss the input data (only based on 4 parameters) and the features that are implicitly getting learnt from the topological representation.

---

### Meta-Review · Program_Chairs · 2020-04-10
**MetaReview of Paper33 by AreaChair1**

**Rating:** 3

**Metareview:**

Multiple reviewers found this a useful contribution on learning distance maps and the differences with prior work seem sufficient. Please in the final version explain the differences with respect to the references mentioned by AnonReviewer4.

**Paper Type:**

methodological development

---

### Decision · Program_Chairs · 2020-04-11

Accept